# Equivalent Statements of Two Multidimensional Hilbert-Type Integral Inequalities with Parameters

**Yiyuan Li [1], Yanru Zhong [2,*] and Bicheng Yang [3]**

[1] School of Art and Design, Guilin University of Electronic Technology, Guilin 541004, China; lyy@guet.edu.cn
[2] School of Computer Science and Information Security, Guilin University of Electronic Technology, Guilin 541004, China
[3] School of Mathematics, Guangdong University of Education, Guangzhou 510303, China; bcyang818@163.com
* Correspondence: 18577399236@163.com

**Abstract:** By means of the weight functions, the idea of introduced parameters and the transfer formulas, two multidimensional Hilbert-type integral inequalities with the general nonhomogeneous kernel as $H(||x||_\alpha^{\lambda_1}||y||_\beta^{\lambda_2})$ $(\lambda_1, \lambda_2 \neq 0)$ are given, which are some extensions of the Hilbert-type integral inequalities in the two-dimensional case. Some equivalent conditions of the best value and several parameters related to the new inequalities are provided. Two corollaries regarding the kernel, represented as $k_\lambda(||x||_\alpha^{\lambda_1}, ||y||_\beta^{\lambda_2})(\lambda_1, \lambda_2 \neq 0)$, are given, and a few new inequalities for the particular parameters are obtained.

**Keywords:** transfer formula; multidimensional Hilbert-type inequality; gamma function; best possible constant factor

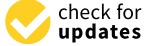



## 1. Introduction

If $0 < \sum_{m=1}^\infty a_m^2 < \infty$ and $0 < \sum_{n=1}^\infty b_n^2 < \infty$, then we have the well-known Hilbert's inequality with the best value $\pi$ as follows (cf. [1], Theorem 315):

$$\sum_{m=1}^\infty \sum_{n=1}^\infty \frac{a_m b_n}{m+n} < \pi \left( \sum_{m=1}^\infty a_m^2 \sum_{n=1}^\infty b_n^2 \right)^{1/2}. \tag{1}$$

Assuming that $0 < \int_0^\infty f^2(x)dx < \infty$ and $0 < \int_0^\infty g^2(y)dy < \infty$, we still have the integral analogue of (1) named in Hilbert's integral inequality as follows (cf. [1], Theorem 316):

$$\int_0^\infty \int_0^\infty \frac{f(x)g(y)}{x+y} dxdy < \pi \left( \int_0^\infty f^2(x)dx \int_0^\infty g^2(y)dy \right)^{1/2}, \tag{2}$$

where $\pi$ is the best value. (1) and (2), with their extensions, played an important role in real analysis. Among them, the paper [2] studied the generalizations of (1) and (2), and the papers [3,4] considered the properties of m-linear Hilbert-type inequality and two kinds of Hilbert-type inequalities involving differential operators.

A half-discrete Hilbert-type inequality was provided in 1934 as follows: If $K(x)(x > 0)$ is decreasing, $p > 1, \frac{1}{p} + \frac{1}{q} = 1, 0 < \varphi(s) = \int_0^\infty K(x)x^{s-1}dx < \infty, f(x) \geq 0$, satisfying

$$0 < \int_0^\infty f^p(x)dx < \infty,$$

then (cf. [1], Theorem 351)

$$\sum_{n=1}^{\infty} n^{p-2} \left( \int_0^{\infty} K(nx) f(x) dx \right)^p < \varphi^p \left( \frac{1}{q} \right) \int_0^{\infty} f^p(x) dx. \tag{3}$$

Some new generalizations and applications of (3) were provided by [5,6] in recent years.

In 2006, by means of the summation formula, Krnic et al. [7] gave a generalization of (1) with the kernel as $\frac{1}{(m+n)^{\lambda}} (0 < \lambda \leq 4)$. In 2019, following [7], Adiyasuren et al. [8] gave a generalization of (1) involving two partial sums. In 2016-2017, Hong et al. [9,10] obtained some equivalent statements of the generalizations of (1) and (2) with the best values related to a few parameters. Two similar results were provided by [11,12]. Among them, the paper [11] considered multidimensional Hardy-type inequalities in Hölder spaces, and the paper [12] studied a new form of Hilbert's integral inequality. To further understand the theory of this field and cite some useful related papers, please see Yang's book [13]. Recently, Hong et al. [14] gave a new half-discrete multidimensional inequality involving one multiple upper limit function as an application.

In this article, following the idea of [7,8], by means of real analysis, the way of introduced parameters and the transfer formulas, two new multidimensional Hilbert-type integral inequalities with the nonhomogeneous kernel as $H(||x||_{\alpha}^{\lambda_1} ||y||_{\beta}^{\lambda_2})(\lambda_1, \lambda_2 \neq 0)$ are given, which are some new extensions of the Hilbert-type integral inequalities in the two-dimensional case. Some equivalent statements of the best possible constant factor and a few parameters related to the new inequalities are provided. Furthermore, two corollaries regard the kernel, represented as $k_{\lambda}(||x||_{\alpha}^{\lambda_1}, ||y||_{\beta}^{\lambda_2})(\lambda_1, \lambda_2 \neq 0)$, are considered, and some new inequalities in a few particular parameters are obtained.

## 2. Some Lemmas

In what follows, we assume that $i_0, j_0 \in \mathbf{N} := \{1, 2, \cdots\}, p > 1, \frac{1}{p} + \frac{1}{q} = 1$, $\sigma_1, \sigma \in \mathbf{R} := (-\infty, \infty), \hat{\sigma} := \frac{\sigma_1}{p} + \frac{\sigma}{q}, \lambda_1, \lambda_2 \neq 0, \alpha, \beta \in \mathbf{R}_+ := (0, \infty)$,

$$||x||_{\alpha} := \left( \sum_{i=1}^{i_0} |x_i|^{\alpha} \right)^{\frac{1}{\alpha}} (x = (x_1, \cdots, x_{i_0}) \in \mathbf{R}^{i_0}),$$

$$||y||_{\beta} := \left( \sum_{j=1}^{j_0} |y_j|^{\beta} \right)^{\frac{1}{\beta}} (y = (y_1, \cdots, y_{j_0}) \in \mathbf{R}^{j_0}).$$

Two functions $f(x), g(y) \geq 0$, satisfying

$$0 < \int_{\mathbf{R}_+^{i_0}} ||x||_{\alpha}^{p(i_0 - \lambda_1 \hat{\sigma}) - i_0} f^p(x) dx < 0 \ and \ 0 < \int_{\mathbf{R}_+^{j_0}} ||y||_{\beta}^{q(j_0 - \lambda_2 \hat{\sigma}) - j_0} g^q(y) dy < \infty.$$

We also suppose that $H(u)$ is a nonnegative measurable function in $\mathbf{R}_+$, such that for any $\eta \in \mathbf{R}$,

$$K(\eta) := \int_0^{\infty} H(u) u^{\eta - 1} du > 0,$$

which means that there exists a positive constant $T > 1$, satisfying $\int_0^T H(u) u^{\eta - 1} du > 0$.

If $M > 0, \psi(u) \ (u > 0)$ is a nonnegative measurable function, then the following transfer formula was provided (cf. [2], (9.3.3)):

$$\int \cdots \int_{\{x \in \mathbf{R}_+^{i_0}; 0 < \sum_{i=1}^{i_0} (\frac{x_i}{M})^{\alpha} \leq 1\}} \psi \left( \sum_{i=1}^{i_0} \left( \frac{x_i}{M} \right)^{\alpha} \right) dx_1 \cdots dx_{i_0} = \frac{M^{i_0} \Gamma^{i_0}(\frac{1}{\alpha})}{\alpha^{i_0} \Gamma(\frac{i_0}{\alpha})} \int_0^1 \psi(u) u^{\frac{i_0}{\alpha} - 1} du. \tag{4}$$

In particular, (i) in view of $||x||_\alpha = M[\sum_{i=1}^{i_0} (\frac{x_i}{M})^\alpha]^{\frac{1}{\alpha}}$, by (4), we have

$$\int_{R_+^{i_0}} \varphi(||x||_\alpha)dx = \lim_{M\to\infty} \int\cdots\int_{\{x\in R_+^{i_0};0<\sum_{i=1}^{i_0}(\frac{x_i}{M})^\alpha\leq 1\}} \varphi(M[\sum_{i=1}^{i_0}(\frac{x_i}{M})^\alpha]^{\frac{1}{\alpha}})dx_1\cdots dx_{i_0}$$

$$= \lim_{M\to\infty} \frac{M^{i_0}\Gamma^{i_0}(\frac{1}{\alpha})}{\alpha^{i_0}\Gamma(\frac{i_0}{\alpha})} \int_0^1 \varphi(Mu^{\frac{1}{\alpha}})u^{\frac{i_0}{\alpha}-1}du \overset{v=Mu^{\frac{1}{\alpha}}}{=\!=\!=} \frac{\Gamma^{i_0}(\frac{1}{\alpha})}{\alpha^{i_0-1}\Gamma(\frac{i_0}{\alpha})} \int_0^\infty \varphi(v)v^{i_0-1}dv; \tag{5}$$

(ii) for $\psi(u) = \varphi(Mu^{\frac{1}{\alpha}}) = 0. u > \frac{1}{M^\alpha}$, by (4), we find

$$\int_{\{x\in R_+^{i_0},||x||_\alpha\leq 1\}} \varphi(||x||_\alpha)dx = \frac{M^{i_0}\Gamma^{i_0}(\frac{1}{\alpha})}{\alpha^{i_0}\Gamma(\frac{i_0}{\alpha})} \int_0^{\frac{1}{M^\alpha}} \varphi(Mu^{\frac{1}{\alpha}})u^{\frac{i_0}{\alpha}-1}du = \frac{\Gamma^{i_0}(\frac{1}{\alpha})}{\alpha^{i_0-1}\Gamma(\frac{i_0}{\alpha})} \int_0^1 \varphi(v)v^{i_0-1}dv; \tag{6}$$

(iii) for $\psi(u) = \varphi(Mu^{\frac{1}{\alpha}}) = 0. u < \frac{1}{M^\alpha}$, by (4), we have

$$\int_{\{x\in R_+^{i_0},||x||_\alpha\geq 1\}} \varphi(||x||_\alpha)dx = \lim_{M\to\infty} \frac{M^{i_0}\Gamma^{i_0}(\frac{1}{\alpha})}{\alpha^{i_0}\Gamma(\frac{i_0}{\alpha})} \int_{\frac{1}{M^\alpha}}^1 \varphi(Mu^{\frac{1}{\alpha}})u^{\frac{i_0}{\alpha}-1}du = \frac{\Gamma^{i_0}(\frac{1}{\alpha})}{\alpha^{i_0-1}\Gamma(\frac{i_0}{\alpha})} \int_1^\infty \varphi(v)v^{i_0-1}dv. \tag{7}$$

For given the main results, we obtain the following weight functions:

**Lemma 1.** Setting $L_\alpha^{(i_0)} := \frac{\Gamma^{i_0}(1/\alpha)}{\alpha^{i_0-1}\Gamma(i_0/\alpha)}$ and $L_\beta^{(j_0)} := \frac{\Gamma^{j_0}(1/\beta)}{\beta^{j_0-1}\Gamma(j_0/\beta)}$, we have the following expressions of the weight functions:

$$\omega(\sigma,y) := \int_{R_+^{i_0}} H(||x||_\alpha^{\lambda_1}||y||_\beta^{\lambda_2})||x||_\alpha^{\lambda_1\sigma-i_0}dx = L_\alpha^{(i_0)} \frac{K(\sigma)}{|\lambda_1|}||y||_\beta^{-\lambda_2\sigma} (\text{ } y \in R_+^{j_0}), \tag{8}$$

$$\varpi(\sigma_1,x) := \int_{R_+^{j_0}} H(||x||_\alpha^{\lambda_1}||y||_\beta^{\lambda_2})||y||_\beta^{\lambda_2\sigma_1-j_0}dy = L_\beta^{(j_0)} \frac{K(\sigma_1)}{|\lambda_2|}||x||_\alpha^{-\lambda_1\sigma_1} (\text{ } x \in R_+^{i_0}). \tag{9}$$

**Proof.** By (5), for $M > 0$, we have

$$\omega(\sigma,y) = \lim_{M\to\infty} M^{\lambda_1\sigma-i_0} \int\cdots\int_{\{x\in R_+^{i_0};0<\sum_{i=1}^{i_0}(\frac{x_i}{M})^\alpha\leq 1\}} H(M^{\lambda_1}[\sum_{i=1}^{i_0}(\frac{x_i}{M})^\alpha]^{\frac{\lambda_1}{\alpha}}||y||_\beta^{\lambda_2})$$

$$\times [\sum_{i=1}^{i_0}(\frac{x_i}{M})^\alpha]^{\frac{\lambda_1\sigma-i_0}{\alpha}} dx_1\cdots dx_{i_0}$$

$$= \lim_{M\to\infty} \frac{M^{i_0}\Gamma^{i_0}(1/\alpha)}{\alpha^{i_0}\Gamma(i_0/\alpha)} M^{\lambda_1\sigma-i_0} \int_0^1 H(M^{\lambda_1}u^{\frac{\lambda_1}{\alpha}}||y||_\beta^{\lambda_2})u^{\frac{\lambda_1\sigma-i_0}{\alpha}} u^{\frac{i_0}{\alpha}-1}du$$

$$= \lim_{M\to\infty} \frac{\Gamma^{i_0}(1/\alpha)}{\alpha^{i_0}\Gamma(i_0/\alpha)} M^{\lambda_1\sigma} \int_0^1 H(M^{\lambda_1}||y||_\beta^{\lambda_2}u^{\frac{\lambda_1}{\alpha}})u^{\frac{\lambda_1\sigma}{\alpha}-1}du. \tag{10}$$

Setting $v = M^{\lambda_1}||y||_\beta^{\lambda_2}u^{\frac{\lambda_1}{\alpha}}$ in the above integral, for $\lambda_1 > 0$, we obtain

$$\omega(\sigma,y) = \frac{\Gamma^{i_0}(1/\alpha)}{|\lambda_1|\alpha^{i_0-1}\Gamma(i_0/\alpha)}||y||_\beta^{-\lambda_2\sigma} \int_0^\infty H(v)v^{\sigma-1}dv,$$

namely, (8) follows. For $\lambda_1 < 0$, by (10), we still can obtain (8). In the same way, for $\lambda_2 \neq 0$, we obtain (9).

This proves the lemma. $\square$

**Lemma 2.** For $b \in \mathbb{R}$, we have the expressions as follows:

$$L_1 := \int_{\{x \in R_+^{i_0}, ||x||_\alpha \leq 1\}} ||x||_\alpha^{b-i_0} dx = \begin{cases} \frac{1}{b} L_\alpha^{(i_0)}, b > 0, \\ \infty, b \leq 0 \end{cases}, \tag{11}$$

$$L_2 := \int_{\{x \in R_+^{i_0}, ||x||_\alpha \geq 1\}} ||x||_\alpha^{-b-i_0} dx = \begin{cases} \frac{1}{b} L_\alpha^{(i_0)}, b > 0, \\ \infty, b \leq 0 \end{cases}. \tag{12}$$

**Proof.** By (6), for $M > 0$, we have

$$L_1 = \int \cdots \int_{\{x \in R_+^{i_0}, \sum_{i=1}^{i_0} (\frac{x_i}{M})^\alpha \leq \frac{1}{M^\alpha}\}} \left[ \sum_{i=1}^{i_0} \left( \frac{x_i}{M} \right)^\alpha \right]^{\frac{b-i_0}{\alpha}} M^{b-i_0} dx_1 \cdots dx_{i_0}$$
$$= \lim_{M \to \infty} \frac{M^{i_0} \Gamma^{i_0}(1/\alpha)}{\alpha^{i_0} \Gamma(i_0/\alpha)} M^{b-i_0} \int_0^{\frac{1}{M^\alpha}} u^{\frac{b-i_0}{\alpha}} u^{\frac{i_0}{\alpha}-1} du = \lim_{M \to \infty} \frac{M^b \Gamma^{i_0}(1/\alpha)}{\alpha^{i_0} \Gamma(i_0/\alpha)} \int_0^{\frac{1}{M^\alpha}} u^{\frac{b}{\alpha}-1} du.$$

For $b > 0$, we find $L_1 = \frac{1}{b} L_\alpha^{(i_0)}$; for $b \leq 0$, it follows that $L_1 = \infty$. Hence, (11) follows. In the same way, by (7), for $M > 0$, we have

$$L_2 = \int \cdots \int_{\{x \in R_+^{i_0}, \sum_{i=1}^{i_0} (\frac{x_i}{M})^\alpha \geq \frac{1}{M^\alpha}\}} \left[ \sum_{i=1}^{i_0} \left( \frac{x_i}{M} \right)^\alpha \right]^{\frac{-b-i_0}{\alpha}} M^{-b-i_0} dx_1 \cdots dx_{i_0}$$
$$= \lim_{M \to \infty} \frac{M^{i_0} \Gamma^{i_0}(1/\alpha)}{\alpha^{i_0} \Gamma(i_0/\alpha)} M^{-b-i_0} \int_{\frac{1}{M^\alpha}}^1 u^{\frac{-b-i_0}{\alpha}} u^{\frac{i_0}{\alpha}-1} du = \lim_{M \to \infty} \frac{M^{-b} \Gamma^{i_0}(1/\alpha)}{\alpha^{i_0} \Gamma(i_0/\alpha)} \int_{\frac{1}{M^\alpha}}^1 u^{\frac{-b}{\alpha}-1} du.$$

For $b > 0$, we find $L_2 = \frac{1}{b} L_\alpha^{(i_0)}$; for $b \leq 0$, it follows that $L_2 = \infty$. Hence, we have (12). This proves the lemma. $\square$

In view of (6) and (7), we give the following expressions:

**Lemma 3.** (i) If $\sigma_1 > \sigma$, then for $0 < \varepsilon < \sigma_1 - \sigma$, we have

$$\widetilde{I}_\varepsilon := \varepsilon \int_{\{x \in R_+^{i_0}; ||x||_\alpha^{\lambda_1} \geq 1\}} ||x||_\alpha^{\lambda_1(\sigma_1 - \frac{\varepsilon}{p}) - i_0} \left[ \int_{\{y \in R_+^{j_0}; ||y||_\beta^{\lambda_2} \leq 1\}} H(||x||_\alpha^{\lambda_1} ||y||_\beta^{\lambda_2}) ||y||_\beta^{\lambda_2(\sigma + \frac{\varepsilon}{q}) - j_0} dy \right] dx = \infty; \tag{13}$$

(ii) If $\sigma_1 < \sigma$, then for $0 < \varepsilon < \sigma - \sigma_1$, we have

$$\hat{I}_\varepsilon := \varepsilon \int_{\{y \in R_+^{j_0}; ||y||_\beta^{\lambda_2} \geq 1\}} ||y||_\beta^{\lambda_2(\sigma_1 - \frac{\varepsilon}{q}) - j_0} \left[ \int_{\{x \in R_+^{i_0}; ||x||_\alpha^{\lambda_1} \leq 1\}} H(||x||_\alpha^{\lambda_1} ||y||_\beta^{\lambda_2}) ||x||_\alpha^{\lambda_1(\sigma + \frac{\varepsilon}{p}) - i_0} dy \right] dx = \infty; \tag{14}$$

(iii) If $\sigma_1 = \sigma$ (in (13)), then

$$I_\varepsilon := \varepsilon \int_{\{x \in R_+^{i_0}; ||x||_\alpha^{\lambda_1} \geq 1\}} ||x||_\alpha^{\lambda_1(\sigma - \frac{\varepsilon}{p}) - i_0} \left[ \int_{\{y \in R_+^{j_0}; ||y||_\beta^{\lambda_2} \leq 1\}} H(||x||_\alpha^{\lambda_1} ||y||_\beta^{\lambda_2}) ||y||_\beta^{\lambda_2(\sigma + \frac{\varepsilon}{q}) - j_0} dy \right] dx$$
$$\geq L_\alpha^{(i_0)} L_\beta^{(j_0)} \frac{K(\sigma)}{|\lambda_1 \lambda_2|} + o(1) (\varepsilon \to 0^+). \tag{15}$$

**Proof.** (i) By (6), for $M, \lambda_2 > 0$, we have

$$h(||x||_\alpha^{\lambda_1}): \quad = ||x||_\alpha^{\lambda_1(\sigma_1+\frac{\varepsilon}{q})} \int_{\{y\in R_+^{j_0};||y||_\beta^{\lambda_2}\leq 1\}} H(||x||_\alpha^{\lambda_1}||y||_\beta^{\lambda_2})||y||_\beta^{\lambda_2(\sigma+\frac{\varepsilon}{q})-j_0} dy$$

$$= ||x||_\alpha^{\lambda_1(\sigma_1+\frac{\varepsilon}{q})} \int\cdots\int_{\{y\in R_+^{j_0};0<\sum_{j=1}^{j_0}(\frac{y_j}{M})^\beta\leq M^{-\beta}\}} H(||x||_\alpha^{\lambda_1}M^{\lambda_2}[\sum_{j=1}^{j_0}(\frac{y_j}{M})^\beta]^{\frac{\lambda_2}{\beta}})$$

$$\times M^{\lambda_2(\sigma+\frac{\varepsilon}{q})-j_0}[\sum_{j=1}^{j_0}(\frac{y_j}{M})^\beta]^{\frac{1}{\beta}[\lambda_2(\sigma+\frac{\varepsilon}{q})-j_0]} dy_1\cdots dy_{j_0}$$

$$= ||x||_\alpha^{\lambda_1(\sigma_1+\frac{\varepsilon}{q})} \lim_{M\to\infty} \frac{M^{j_0}\Gamma^{j_0}(1/\beta)}{\beta^{j_0}\Gamma(j_0/\beta)} \int_0^{M^{-\beta}} H(||x||_\alpha^{\lambda_1}M^{\lambda_2}u^{\frac{\lambda_2}{\beta}})M^{\lambda_2(\sigma+\frac{\varepsilon}{q})-j_0}u^{\frac{1}{\beta}[\lambda_2(\sigma+\frac{\varepsilon}{q})-j_0]}u^{\frac{j_0}{\beta}-1}du$$

$$= ||x||_\alpha^{\lambda_1(\sigma_1+\frac{\varepsilon}{q})} \lim_{M\to\infty} \frac{M^{\lambda(\sigma+\frac{\varepsilon}{q})_2}\Gamma^{j_0}(1/\beta)}{\beta^{j_0}\Gamma(j_0/\beta)} \int_0^{M^{-\beta}} H(||x||_\alpha^{\lambda_1}M^{\lambda_2}u^{\frac{\lambda_2}{\beta}})u^{\frac{1}{\beta}\lambda_2(\sigma+\frac{\varepsilon}{q})-1}du.$$

Setting $v=||x||_\alpha^{\lambda_1}M^{\lambda_2}u^{\frac{\lambda_2}{\beta}}$ in the above expression, in view of $\lambda_2>0$, it follows that

$$h(||x||_\alpha^{\lambda_1}) = \frac{1}{\lambda_2}L_\beta^{(j_0)}||x||_\alpha^{\lambda_1(\sigma_1-\sigma)} \int_0^{||x||_\alpha^{\lambda_1}} H(v)v^{(\sigma+\frac{\varepsilon}{q})-1}dv. \tag{16}$$

For $\lambda_2<0$, by (7), we obtain

$$h(||x||_\alpha^{\lambda_1}) = ||x||_\alpha^{\lambda_1(\sigma+1\frac{\varepsilon}{q})} \int_{\{y\in R_+^{j_0};||y||_\beta\geq 1\}} H(||x||_\alpha^{\lambda_1}||y||_\beta^{\lambda_2})||y||_\beta^{\lambda_2(\sigma+\frac{\varepsilon}{q})-j_0}dy$$

$$= ||x||_\alpha^{\lambda_1(\sigma_1+\frac{\varepsilon}{q})} \int\cdots\int_{\{y\in R_+^{j_0};\sum_{j=1}^{j_0}(\frac{y_j}{M})^\beta\geq M^{-\beta}\}} H(||x||_\alpha^{\lambda_1}M^{\lambda_2}[\sum_{j=1}^{j_0}(\frac{y_j}{M})^\beta]^{\frac{\lambda_2}{\beta}})$$

$$\times M^{\lambda_2(\sigma+\frac{\varepsilon}{q})-j_0}[\sum_{j=1}^{j_0}(\frac{y_j}{M})^\beta]^{\frac{1}{\beta}[\lambda_2(\sigma+\frac{\varepsilon}{q})-j_0]} dy_1\cdots dy_{j_0}$$

$$= ||x||_\alpha^{\lambda_1(\sigma_1+\frac{\varepsilon}{q})} \lim_{M\to\infty} \frac{M^{j_0}\Gamma^{j_0}(1/\beta)}{\beta^{j_0}\Gamma(j_0/\beta)} \int_{M^{-\beta}}^1 H(||x||_\alpha^{\lambda_1}M^{\lambda_2}u^{\frac{\lambda_2}{\beta}})M^{\lambda_2(\sigma+\frac{\varepsilon}{q})-j_0}u^{\frac{1}{\beta}[\lambda_2(\sigma+\frac{\varepsilon}{q})-j_0]}u^{\frac{j_0}{\beta}-1}du$$

$$= ||x||_\alpha^{\lambda_1(\sigma_1+\frac{\varepsilon}{q})} \lim_{M\to\infty} \frac{M^{\lambda(\sigma+\frac{\varepsilon}{q})^2}\Gamma^{j_0}(1/\beta)}{\beta^{j_0}\Gamma(j_0/\beta)} \int_{M^{-\beta}}^1 H(||x||_\alpha^{\lambda_1}M^{\lambda_2}u^{\frac{\lambda_2}{\beta}})u^{\frac{1}{\beta}\lambda_2(\sigma+\frac{\varepsilon}{q})-1}du.$$

Setting $v=||x||_\alpha^{\lambda_1}M^{\lambda_2}u^{\frac{\lambda_2}{\beta}}$ in the above expression, in view of $\lambda_2<0$, it follows that

$$h(||x||_\alpha^{\lambda_1}) = \frac{1}{-\lambda_2}L_\beta^{(j_0)}||x||_\alpha^{\lambda_1(\sigma_1-\sigma)} \int_0^{||x||_\alpha^{\lambda_1}} H(v)v^{(\sigma+\frac{\varepsilon}{q})-1}dv. \tag{17}$$

In view of (16) and (17), we have

$$\widetilde{I}_\varepsilon = \varepsilon \int_{\{x\in R_+^{i_0};||x||_\alpha^{\lambda_1}\geq 1\}} ||x||_\alpha^{-\lambda_1\varepsilon-i_0}h(||x||_\alpha^{\lambda_1})dx$$

$$= \frac{\varepsilon}{|\lambda_2|}L_\beta^{(j_0)} \int_{\{x\in R_+^{i_0};||x||_\alpha^{\lambda_1}\geq 1\}} ||x||_\alpha^{-\lambda_1(\sigma-\sigma_1+\varepsilon)-i_0}[\int_0^{||x||_\alpha^{\lambda_1}} H(v)v^{(\sigma+\frac{\varepsilon}{q})-1}dv]dx.$$

For $\lambda_1>0$ by (7), we have for $M>0$ that

$$\widetilde{I}_\varepsilon = \frac{\varepsilon}{|\lambda_2|}L_\beta^{(j_0)} \int\cdots\int_{\{x\in R_+^{i_0};\sum_{i=1}^{i_0}(\frac{x_i}{M})^\alpha\geq M^{-\alpha}\}} M^{-\lambda_1(\sigma-\sigma_1+\varepsilon)-i_0}[\sum_{i=1}^{i_0}(\frac{x_i}{M})^\alpha]^{\frac{-\lambda_1(\sigma-\sigma_1+\varepsilon)-i_0}{\alpha}}$$

$$\times [\int_0^{M^{\lambda_1}[\sum_{i=1}^{i_0}(\frac{x_i}{M})^\alpha]^{\frac{\lambda_1}{\alpha}}} H(v)v^{(\sigma+\frac{\varepsilon}{q})-1}dv]dx_1\cdots dx_{i_0}$$

$$= \frac{\varepsilon}{|\lambda_2|}L_\beta^{(j_0)} \lim_{M\to\infty} \frac{M^{i_0}\Gamma^{i_0}(1/\alpha)}{\alpha^{i_0}\Gamma(i_0/\alpha)} \int_{M^{-\alpha}}^1 M^{-\lambda_1(\sigma-\sigma_1+\varepsilon)-i_0}u^{\frac{-\lambda_1(\sigma-\sigma_1+\varepsilon)-i_0}{\alpha}}[\int_0^{M^{\lambda_1}u^{\frac{\lambda_1}{\alpha}}} H(v)v^{(\sigma+\frac{\varepsilon}{q})-1}dv]u^{\frac{i_0}{\alpha}-1}du$$

$$= \frac{\varepsilon}{|\lambda_2|}L_\beta^{(j_0)} \lim_{M\to\infty} \frac{M^{-\lambda_1(\sigma-\sigma_1+\varepsilon)-i_0}(1/\alpha)}{\alpha^{i_0}\Gamma(i_0/\alpha)} \int_{M^{-\alpha}}^1 u^{\frac{-\lambda_1(\sigma-\sigma_1+\varepsilon)}{\alpha}-1}[\int_0^{M^{\lambda_1}u^{\frac{\lambda_1}{\alpha}}} H(v)v^{(\sigma+\frac{\varepsilon}{q})-1}dv]du$$

$$\overset{t=M^{\lambda_1}u^{\frac{\lambda_1}{\alpha}}}{=} \frac{\varepsilon}{|\lambda_1\lambda_2|}L_\beta^{(j_0)}L_\alpha^{(i_0)} \int_1^\infty t^{-(\sigma-\sigma_1+\varepsilon)-1}[\int_0^t H(v)v^{(\sigma+\frac{\varepsilon}{q})-1}dv]dt.$$

For $\lambda_1 < 0, M > 0$, by (7), we still have

$$\widetilde{I}_\varepsilon = \frac{\varepsilon}{|\lambda_2|} L_\beta^{(j_0)} \int \cdots \int_{\{x \in R_+^{i_0}; 0 < \sum_{i=1}^{i_0} (\frac{x_i}{M})^\alpha \leq M^{-\alpha}\}} M^{-\lambda_1(\sigma-\sigma_1+\varepsilon)-i_0} \left[\sum_{i=1}^{i_0} \left(\frac{x_i}{M}\right)^\alpha\right]^{\frac{-\lambda_1(\sigma-\sigma_1+\varepsilon)-i_0}{\alpha}}$$

$$\times \left[\int_0^{M^{\lambda_1}[\sum_{i=1}^{i_0}(\frac{x_i}{M})^\alpha]^{\frac{\lambda_1}{\alpha}}} H(v) v^{(\sigma+\frac{\varepsilon}{q})-1} dv\right] dx_1 \cdots dx_{i_0}$$

$$= \frac{\varepsilon}{|\lambda_2|} L_\beta^{(j_0)} \lim_{M \to \infty} \frac{M^{i_0} \Gamma^{i_0}(1/\alpha)}{\alpha^{i_0} \Gamma(i_0/\alpha)} \int_0^{M^{-\alpha}} M^{-\lambda_1(\sigma-\sigma_1+\varepsilon)-i_0} u^{\frac{-\lambda_1(\sigma-\sigma_1+\varepsilon)-i_0}{\alpha}} \left[\int_0^{M^{\lambda_1} u^{\frac{\lambda_1}{\alpha}}} H(v) v^{(\sigma+\frac{\varepsilon}{q})-1} dv\right] u^{\frac{i_0}{\alpha}-1} du$$

$$= \frac{\varepsilon}{|\lambda_2|} L_\beta^{(j_0)} \lim_{M \to \infty} \frac{M^{-\lambda_1(\sigma-\sigma_1+\varepsilon)-i_0}(1/\alpha)}{\alpha^{i_0} \Gamma(i_0/\alpha)} \int_0^{M^{-\alpha}} u^{\frac{-\lambda_1\varepsilon}{\alpha}-1} \left[\int_0^{M^{\lambda_1} u^{\frac{\lambda_1}{\alpha}}} H(v) v^{(\sigma+\frac{\varepsilon}{q})-1} dv\right] du$$

$$\overset{t=M^{\lambda_1} u^{\frac{\lambda_1}{\alpha}}}{=\!=\!=} \frac{\varepsilon}{|\lambda_1 \lambda_2|} L_\beta^{(j_0)} L_\alpha^{(i_0)} \int_1^\infty t^{-(\sigma-\sigma_1+\varepsilon)-1} \left[\int_0^t H(v) v^{(\sigma+\frac{\varepsilon}{q})-1} dv\right] dt.$$

Hence, we have

$$\widetilde{I}_\varepsilon \geq \frac{\varepsilon}{|\lambda_1 \lambda_2|} L_\beta^{(j_0)} L_\alpha^{(i_0)} \int_T^\infty t^{-(\sigma-\sigma_1+\varepsilon)-1} dt \int_0^T H(v) v^{(\sigma+\frac{\varepsilon}{q})-1} dv \, (T > 1), \tag{18}$$

Satisfying $\int_0^T H(v) v^{(\sigma+\frac{\varepsilon}{q})-1} dv > 0$. For $\sigma - \sigma_1 + \varepsilon < 0$, we have $\int_T^\infty t^{-(\sigma-\sigma_1+\varepsilon)-1} dt = \infty$, in view of (18), we have $\widetilde{I}_\varepsilon = \infty$. Hence, we have (13).

(ii) In the same way, by the symmetry, we have (14).

(iii) If $\sigma_1 = \sigma$, then in view of (18), by Fubini theorem and Fatou lemma (cf. [15]), we obtain

$$\lim_{\varepsilon \to 0^+} I_\varepsilon = \lim_{\varepsilon \to 0^+} \frac{\varepsilon}{|\lambda_1 \lambda_2|} L_\beta^{(j_0)} L_\alpha^{(i_0)}$$
$$\times \left\{\int_1^\infty t^{-\varepsilon-1} \left[\int_0^1 H(v) v^{(\sigma+\frac{\varepsilon}{q})-1} dv\right] dt + \int_1^\infty t^{-\varepsilon-1} \left[\int_1^t H(v) v^{(\sigma+\frac{\varepsilon}{q})-1} dv\right] dt\right\}$$

$$= \lim_{\varepsilon \to 0^+} \frac{\varepsilon}{|\lambda_1 \lambda_2|} L_\beta^{(j_0)} L_\alpha^{(i_0)} \left[\frac{1}{\varepsilon} \int_0^1 H(v) v^{(\sigma+\frac{\varepsilon}{q})-1} dv + \int_1^\infty \left(\int_v^\infty t^{-\varepsilon-1} dt\right) H(v) v^{(\sigma+\frac{\varepsilon}{q})-1} dv\right]$$

$$= \frac{1}{|\lambda_1 \lambda_2|} L_\beta^{(j_0)} L_\alpha^{(i_0)} \lim_{\varepsilon \to 0^+} \left[\int_0^1 H(v) v^{(\sigma+\frac{\varepsilon}{q})-1} dv + \int_1^\infty H(v) v^{(\sigma-\frac{\varepsilon}{p})-1} dv\right]$$

$$\geq \frac{1}{|\lambda_1 \lambda_2|} L_\beta^{(j_0)} L_\alpha^{(i_0)} \left(\int_0^1 \lim_{\varepsilon \to 0^+} H(v) v^{\sigma+\frac{\varepsilon}{q}-1} dv + \int_1^\infty \lim_{\varepsilon \to 0^+} H(v) v^{\sigma-\frac{\varepsilon}{p}-1} dv\right) = L_\alpha^{(i_0)} L_\beta^{(j_0)} \frac{K(\sigma)}{|\lambda_1 \lambda_2|},$$

namely, (15) follows.

The lemma is proved. □

By Lemma 1, we obtain the following main inequality:

**Lemma 4.** *If* $K(\eta) \in R_+ (\eta \in \{\sigma_1, \sigma\})$, *then we have the following inequality*

$$I := \int_{R_+^{i_0}} \int_{R_+^{j_0}} H(||x||_\alpha^{\lambda_1} ||y||_\beta^{\lambda_2}) f(x) g(y) dx dy < \left(L_\beta^{(j_0)} \frac{K(\sigma_1)}{|\lambda_2|}\right)^{\frac{1}{p}} \left(L_\alpha^{(i_0)} \frac{K(\sigma)}{|\lambda_1|}\right)^{\frac{1}{q}}$$
$$\times \left[\int_{R_+^{i_0}} ||x||_\alpha^{p(i_0-\lambda_1\hat{\sigma})-i_0} f^p(x) dx\right]^{\frac{1}{p}} \left[\int_{R_+^{j_0}} ||y||_\beta^{q(j_0-\lambda_2\hat{\sigma})-j_0} g^q(y) dy\right]^{\frac{1}{q}}. \tag{19}$$

**Proof.** By Hölder's inequality (cf. [16]), we have

$$I = \int_{R_+^{i_0}} \int_{R_+^{j_0}} H(||x||_\alpha^{\lambda_1} ||y||_\beta^{\lambda_2}) \left[\frac{||y||_\beta^{(\lambda_2\sigma_1-j_0)/p}}{||x||_\alpha^{(\lambda_1\sigma-i_0)/q}} f(x)\right] \left[\frac{||x||_\alpha^{(\lambda_1\sigma-i_0)/q}}{||y||_\beta^{(\lambda_2\sigma_1-j_0)/p}} g(y)\right] dx dy$$

$$\leq \left\{\int_{R_+^{i_0}} \left[\int_{R_+^{j_0}} H(||x||_\alpha^{\lambda_1} ||y||_\beta^{\lambda_2}) \frac{||y||_\beta^{\lambda_2\sigma_1-j_0}}{||x||_\alpha^{(\lambda_1\sigma-i_0)(p-1)}} dy\right] f^p(x) dx\right\}^{\frac{1}{p}}$$

$$\times \{\int_{R_+^{j_0}} [\int_{R_+^{i_0}} H(||x||_\alpha^{\lambda_1} ||y||_\beta^{\lambda_2}) \frac{||x||_\alpha^{\lambda_1\sigma-i_0}}{||y||_\beta^{(\lambda_2\sigma_1-j_0)(q-1)}} dx] g^q(y) dy\}^{\frac{1}{q}}$$

$$= [\int_{R_+^{i_0}} \omega(\sigma_1, x) ||x||_\alpha^{(p-1)(i_0-\lambda_1\sigma)} f^p(x) dx]^{\frac{1}{p}}$$

$$\times [\int_{R_+^{j_0}} \omega(\sigma, y) ||y||_\beta^{(q-1)(j_0-\lambda_2\sigma_1)} g^q(y) dy]^{\frac{1}{q}}. \tag{20}$$

If (20) pertain to the form of equality, then (cf. [16]), there exist constants $A$ and $B$, satisfying they are not both zero, and

$$A \frac{||y||_\beta^{\lambda_2\sigma_1-j_0}}{||x||_\alpha^{(\lambda_1\sigma-i_0)(p-1)}} f^p(x) = B \frac{||x||_\alpha^{\lambda_1\sigma-i_0}}{||y||_\beta^{(\lambda_2\sigma_1-j_0)(q-1)}} g^q(y) a.e. in R_+^{i_0} \times R_+^{j_0}.$$

Assuming that $A \neq 0$, there exists a $y \in R_+^{j_0}$, such that

$$||x||_\alpha^{p(i-0\lambda_1\hat{\sigma})-i_0} f^p(x) = \frac{Bg^q(y)}{A||y||_\beta^{q(\lambda_2\sigma_1-j_0)}} ||x||_\alpha^{\lambda_1(\sigma-\sigma_1)-i_0} a.e. in R_+^{i_0},$$

which contradicts that

$$0 < \int_{R_+^{i_0}} ||x||_\alpha^{p[i_0-\lambda_1\hat{\sigma}]-i_0} f^p(x) dx < \infty.$$

In fact, by (11) and (12), for $b = \lambda_1(\sigma - \sigma_1) \in R$, we have

$$\int_{R_+^{i_0}} ||x||_\alpha^{b-i_0} dx = \int_{\{x \in R_+^{i_0}; ||x||_\alpha \leq 1\}} ||x||_\alpha^{b-i_0} dx + \int_{\{x \in R_+^{i_0}; ||x||_\alpha \geq 1\}} ||x||_\alpha^{b-i_0} dx = \infty.$$

By (8) and (9), we obtain (19).
This proves the lemma. □

**Remark 1** (i) In particular, for $\sigma = 1\sigma$ in (19), we have $\hat{\sigma} = \sigma$,

$$0 < \int_{R_+^{i_0}} ||x||_\alpha^{p(i_0-\lambda_1\sigma)-i_0} f^p(x) dx < \infty, 0 < \int_{R_+^{j_0}} ||y||_\beta^{q(j_0-\lambda_2\sigma)-j_0} g^q(y) dy < \infty,$$

and the following:

$$I = \int_{R_+^{j_0}} \int_{R_+^{i_0}} H(||x||_\alpha^{\lambda_1} ||y||_\beta^{\lambda_2}) f(x) g(y) dx dy < (L_\beta^{(j_0)} \frac{1}{|\lambda_2|})^{\frac{1}{p}} (L_\alpha^{(i_0)} \frac{1}{|\lambda_1|})^{\frac{1}{q}} K(\sigma)$$

$$\times [\int_{R_+^{i_0}} ||x||_\alpha^{p(i_0-\lambda_1\sigma)-i_0} f^p(x) dx]^{\frac{1}{p}} [\int_{R_+^{j_0}} ||y||_\beta^{q(j_0-\lambda_2\sigma)-j_0} g^q(y) dy]^{\frac{1}{q}}. \tag{21}$$

(ii) By Hölder's inequality (cf. [16]), we still have

$$0 < K(\hat{\sigma}) = K(\frac{\sigma_1}{p} + \frac{\sigma}{q}) = \int_0^\infty H(u) u^{\frac{\sigma_1}{p}+\frac{\sigma}{q}-1} du$$

$$= \int_0^\infty H(u)(u^{\frac{\sigma_1-1}{p}})(u^{\frac{\sigma-1}{q}}) du$$

$$\leq (\int_0^\infty H(u) u^{\sigma_1-1} du)^{\frac{1}{p}} (\int_0^\infty H(u) u^{\sigma-1} du)^{\frac{1}{q}} = (K(\sigma_1))^{\frac{1}{p}} (K(\sigma))^{\frac{1}{q}} < \infty. \tag{22}$$

Now, we use Lemmas 2 and 3 to show the best value in the key inequality (21).

**Lemma 5.** For $K(\sigma) \in R_+$, $(L_\beta^{(j_0)} \frac{1}{|\lambda_2|})^{\frac{1}{p}} (L_\alpha^{(i_0)} \frac{1}{|\lambda_1|})^{\frac{1}{q}} K(\sigma)$ in (21) is the best value.

**Proof.** For any $\varepsilon > 0$, we set

$$f_\varepsilon(x) := \begin{cases} 0, ||x||_\alpha^{\lambda_1} < 1, \\ ||x||_\alpha^{\lambda_1(\sigma - \frac{\varepsilon}{p}) - i_0}, ||x||_\alpha^{\lambda_1} \geq 1, \end{cases} \quad g_\varepsilon(y) := \begin{cases} ||y||_\beta^{\lambda_2(\sigma + \frac{\varepsilon}{q}) - j_0}, ||y||_\beta^{\lambda_2} \leq 1, \\ 0, ||y||_\beta^{\lambda_2} > 1. \end{cases}$$

By (11) and (12), we have

$$\int_{x \in R_+^{i_0}} ||x||_\alpha^{p(i_0 - \lambda_1\sigma) - i_0} f_\varepsilon^p(x) dx = \int_{\{x \in R_+^{i_0}; ||x||_\alpha^{\lambda_1} \geq 1\}} ||x||_\alpha^{-\lambda_1\varepsilon - i_0} dx$$

$$= \begin{cases} \int_{\{x \in R_+^{i_0}; ||x||_\alpha \leq 1\}} ||x||_\alpha^{|\lambda_1|\varepsilon - i_0} dx, \lambda_1 < 0 \\ \int_{\{x \in R_+^{i_0}; ||x||_\delta \geq 1\}} ||x||_\alpha^{-|\lambda_1|\varepsilon - i_0} dx, \lambda_1 > 0 \end{cases} = \frac{1}{|\lambda_1|\varepsilon} L_\alpha^{(i_0)},$$

$$\int_{R_+^{j_0}} ||y||_\beta^{q(j_0 - \sigma) - j_0} g_\varepsilon^q(y) dy = \int_{\{y \in R_+^{j_0}; ||y||_\beta^{\lambda_2} \leq 1\}} ||y||_\beta^{\lambda_2\varepsilon - j_0} dy = \frac{1}{|\lambda_2|\varepsilon} L_\beta^{(j_0)}. \tag{23}$$

If there exists a positive constant

$$M \leq (L_\beta^{(j_0)} \frac{1}{|\lambda_2|})^{\frac{1}{p}} (L_\alpha^{(i_0)} \frac{1}{|\lambda_1|})^{\frac{1}{q}} K(\sigma),$$

such that (21) is valid as we replace $(L_\beta^{(j_0)} \frac{1}{|\lambda_2|})^{\frac{1}{p}} (L_\alpha^{(i_0)} \frac{1}{|\lambda_1|})^{\frac{1}{q}} K(\sigma)$ by $M$, then in particular, by (15), we have

$$L_\alpha^{(i_0)} L_\beta^{(j_0)} \frac{K(\sigma)}{|\lambda_1\lambda_2|} + o(1) \leq I_\varepsilon = \varepsilon \int_{R_+^{j_0}} \int_{R_+^{i_0}} H(||x||_\alpha^{\lambda_1} ||y||_\beta^{\lambda_2}) f_\varepsilon(x) g_\varepsilon(y) \varepsilon dx dy$$

$$< \varepsilon M [\int_{R_+^{i_0}} ||x||_\alpha^{p(i_0 - \lambda_1\sigma) - i_0} f_\varepsilon{}^p(x) dx]^{\frac{1}{p}} [\int_{R_+^{j_0}} ||y||_\beta^{q(j_0 - \lambda_2\sigma) - j_0} g_\varepsilon{}^q(y) dy]^{\frac{1}{q}}$$

$$= M (L_\alpha^{(i_0)} \frac{1}{|\lambda_1|})^{\frac{1}{p}} (L_\beta^{(j_0)} \frac{1}{|\lambda_2|})^{\frac{1}{q}}.$$

For $\varepsilon \to 0^+$, it follows that

$$L_\alpha^{(i_0)} L_\beta^{(j_0)} \frac{K(\sigma)}{|\lambda_1\lambda_2|} \leq M (L_\alpha^{(i_0)} \frac{1}{|\lambda_1|})^{\frac{1}{p}} (L_\beta^{(j_0)} \frac{1}{|\lambda_2|})^{\frac{1}{q}}.$$

We find that $(L_\beta^{(j_0)} \frac{1}{|\lambda_2|})^{\frac{1}{p}} (L_\alpha^{(i_0)} \frac{1}{|\lambda_1|})^{\frac{1}{q}} K(\sigma) \leq M$, which follows that

$$M = (L_\beta^{(j_0)} \frac{1}{|\lambda_2|})^{\frac{1}{p}} (L_\alpha^{(j_0)} \frac{1}{|\lambda_1|})^{\frac{1}{q}} K(\sigma)$$

is the best possible constant factor of (21).
This proves the lemma. □

## 3. Main Results and Two Corollaries

**Theorem 1.** For $p > 1, \frac{1}{p} + \frac{1}{q} = 1$, if there exists $M(\geq 0)$, satisfying the following inequality holds:

$$I = \int_{R_+^{j_0}} \int_{R_+^{i_0}} H(||x||_\alpha^{\lambda_1} ||y||_\beta^{\lambda_2}) f(x) g(y) dx dy$$

$$\leq M [\int_{R_+^{i_0}} ||x||_\alpha^{p(i_0 - \lambda_1\sigma_1) - i_0} f^p(x) dx]^{\frac{1}{p}} [\int_{R_+^{j_0}} ||y||_\beta^{q(j_0 - \lambda_2\sigma) - j_0} g^q(y) dy]^{\frac{1}{q}}, \tag{24}$$

then we have $\sigma_1 = \sigma$ and $M > 0$. Hence, $K(\sigma) \in R_+$ and

$$(L_\beta^{(j_0)} \frac{1}{|\lambda_2|})^{\frac{1}{p}} (L_\alpha^{(i_0)} \frac{1}{|\lambda_1|})^{\frac{1}{q}} K(\sigma) (\in (0, M])$$

is the best value of (24) (for $\sigma_1 = \sigma$).

**Proof.** If $\sigma_1 < \sigma$, then for any $\varepsilon > 0$, we set

$$\widetilde{f}_\varepsilon(x) := \begin{cases} 0, ||x||_\alpha^{\lambda_1} < 1, \\ ||x||_\alpha^{\lambda(\sigma_1 - \frac{\varepsilon}{p})1 - i_0}, ||x||_\alpha^{\lambda_1} \geq 1, \end{cases}$$

$$\widetilde{g}_\varepsilon(y) := \begin{cases} ||y||_\beta^{\lambda_2(\sigma + \frac{\varepsilon}{q}) - j_0}, ||y||_\beta^{\lambda_2} \leq 1, \\ 0, ||y||_\beta^{\lambda_2} > 1. \end{cases}$$

By (8), (18) and (19), we have

$$\infty = \widetilde{I}_\varepsilon = \varepsilon \int_{R_+^{j_0}} \int_{R_+^{i_0}} H(||x||_\alpha^{\lambda_1} ||y||_\beta^{\lambda_2}) \widetilde{f}_\varepsilon(x) \widetilde{g}_\varepsilon(y) dx dy$$

$$\leq \varepsilon M [\int_{R_+^{i_0}} ||x||_\alpha^{p(i_0 - \lambda_1\sigma_1) - i_0} \widetilde{f}_\varepsilon^p(x) dx]^{\frac{1}{p}} [\int_{R_+^{j_0}} ||y||_\beta^{q(j_0 - \lambda_2\sigma) - j_0} \widetilde{g}_\varepsilon^q(y) dy]^{\frac{1}{q}}$$

$$= \varepsilon M (\int_{\{x \in R_+^{i_0}; ||x||_\alpha^{\lambda_1} \geq 1\}} ||x||_\alpha^{-\lambda_1\varepsilon - i_0} dx)^{\frac{1}{p}} (\int_{\{y \in R_+^{j_0}; ||y||_\beta^{\lambda_2} \leq 1\}} ||y||_\beta^{\lambda_2\varepsilon - j_0} dy)^{\frac{1}{q}} < \infty,$$

which is a contradiction.

If $\sigma_1 < \sigma$, then for any $\varepsilon > 0$, we set

$$\hat{f}_\varepsilon(x) := \begin{cases} ||x||_\alpha^{\lambda_1(\sigma_1 + \frac{\varepsilon}{p}) - i_0}, ||x||_\alpha^{\lambda_1} \leq 1, \\ 0, ||x||_\alpha^{\lambda_1} > 1, \end{cases}$$

$$\hat{g}_\varepsilon(y) := \begin{cases} 0, ||y||_\beta^{\lambda_2} < 1, \\ ||y||_\beta^{\lambda_2(\sigma - \frac{\varepsilon}{q}) - j_0}, ||y||_\beta^{\lambda_2} \geq 1. \end{cases}$$

By (9), (18) and (19), in the same way, we still obtain a contradiction.

Hence, we have $\sigma_1 = \sigma$.

For $\sigma_1 = \sigma$ in (19), replacing $f(x)(resp.g(y))$ by $f_\varepsilon(x)(resp.g_\varepsilon(y))$ in Lemma 5 and following the proof of Lemma 5, for $K(\sigma) > 0$, we still find

$$0 < (L_\beta^{(j_0)} \frac{1}{|\lambda_2|})^{\frac{1}{p}} (L_\alpha^{(i_0)} \frac{1}{|\lambda_1|})^{\frac{1}{q}} K(\sigma) \leq M < \infty,$$

which follows that

$$0 < K(\sigma) \leq M (L_\beta^{(j_0)} \frac{1}{|\lambda_2|})^{-\frac{1}{p}} (L_\alpha^{(i_0)} \frac{1}{|\lambda_1|})^{-\frac{1}{q}} < \infty.$$

By Lemma 5, $(L_\beta^{(j_0)} \frac{1}{|\lambda_2|})^{\frac{1}{p}} (L_\alpha^{(i_0)} \frac{1}{|\lambda_1|})^{\frac{1}{q}} K(\sigma) (\in (0, M])$ is the best possible constant of (24) (for $\sigma_1 = \sigma$).

This proves the theorem. $\square$

**Theorem 2.** For $K(\eta) \in R_+ (\eta \in \{\sigma, \sigma_1\})$, we have the following equivalent statements:

(i) Both $(K(\sigma_1))^{\frac{1}{p}}(K(\sigma))^{\frac{1}{q}}$ and $K(\frac{\sigma_1}{p} + \frac{\sigma}{q})$ are independent of $p, q$;

$$(ii) \ (K(\sigma_1))^{\frac{1}{p}}(K(\sigma))^{\frac{1}{q}} \leq K(\frac{\sigma_1}{p} + \frac{\sigma}{q}); \tag{25}$$

(iii) $\sigma_1 = \sigma$;

(iv) The constant factor $(L_\beta^{(j_0)} \frac{K(\sigma_1)}{|\lambda_2|})^{\frac{1}{p}} (L_\alpha^{(i_0)} \frac{K(\sigma)}{|\lambda_1|})^{\frac{1}{q}}$ in (19) is the best value;

(v) there exists a constant $M$, such that (24) holds.

**Proof.** (ii)⇒(iii). By (25), it follows that (22) protains the form of equality. Then, there exist $A$ and $B$, satisfying they are not both zero and $Au^{\sigma_1-1} = Bu^{\sigma-1}$ a.e. in $R_+$ (cf. [16]). Supposing that $A \neq 0$, we have $u^{\sigma_1-\sigma} = \frac{B}{A}$ a.e. in $R_+$, and then $\sigma_1 - \sigma = 0$, namely, $\sigma_1 = \sigma$.

(iii)⇒(iv). In view of Lemma 5, we obtain (iv).

(iv)⇒(ii). If the constant factor $(L_\beta^{(j_0)} \frac{K(\sigma_1)}{|\lambda_2|})^{\frac{1}{p}} (L_\alpha^{(i_0)} \frac{K(\sigma)}{|\lambda_1|})^{\frac{1}{q}}$ in (19) is the best value, then by (21) (for $\sigma = \hat{\sigma}$), we have

$$(L_\beta^{(j_0)} \frac{K(\sigma_1)}{|\lambda_2|})^{\frac{1}{p}} (L_\alpha^{(i_0)} \frac{K(\sigma)}{|\lambda_1|})^{\frac{1}{q}} \leq (L_\beta^{(j_0)} \frac{1}{|\lambda_2|})^{\frac{1}{p}} (L_\alpha^{(i_0)} \frac{1}{|\lambda_1|})^{\frac{1}{q}} K(\hat{\sigma}) \in R_+,$$

namely, (25) follows. Hence, it follows that (ii)⇔(iii)⇔(iv).

(i)⇒(ii). By (i), we find

$$(K(\sigma_1))^{\frac{1}{p}}(K(\sigma))^{\frac{1}{q}} = \lim_{p\to\infty}\lim_{q\to 1}(K(\sigma_1))^{\frac{1}{p}}(K(\sigma))^{\frac{1}{q}} = K(\sigma),$$

and then, in view of Fatou lemma (cf. [15]), we have

$$(K(\sigma_1))^{\frac{1}{p}}(K(\sigma))^{\frac{1}{q}} = K(\sigma) = K(\lim_{p\to\infty}\frac{\sigma_1-\sigma}{p}+\sigma) \leq \lim_{p\to\infty}K(\frac{\sigma_1-\sigma}{p}+\sigma) = K(\frac{\sigma_1}{p}+\frac{\sigma}{q}),$$

namely, (25) follows.

(iii)⇒(i). For $\sigma_1 = \sigma$, both $(K(\sigma_1))^{\frac{1}{p}}(K(\sigma))^{\frac{1}{q}}$ and $K(\frac{\sigma_1}{p} + \frac{\sigma}{q})$ equal $K(\sigma)$, which are independent of $p, q$. Hence, we have (i)⇔(ii)⇔(iii)⇔(iv).

(v)⇒(iii). By Theorem 1, for $K(\eta) \in R_+ (\eta = \sigma, \sigma)1$, we still have $\sigma_1 = \sigma$.

(iii)⇒(v). If $\sigma_1 = \sigma$, then by Lemma 5, we set $M \geq (L_\beta^{(j_0)} \frac{1}{|\lambda_2|})^{\frac{1}{p}} (L_\alpha^{(i_0)} \frac{1}{|\lambda_1|})^{\frac{1}{q}} K(\sigma)$, and then (24) holds. Hence, we have (iii)⇔(v).

Therefore, we have (i)⇔(ii)⇔(iii)⇔(iv)⇔(v).

This proved that theorem. □

Replacing $\lambda_1$ to $-\lambda_1$ in Theorems 1 and 2, setting $H(v) = k_\lambda(1, v)$, where $k_\lambda(u, v)$ is a homogeneous function of degree $-\lambda$, such that $k_\lambda(tu, tv) = t^{-\lambda}k_\lambda(u, v)(t, u, v > 0)$, and

$$K_\lambda(\eta) := \int_0^\infty k_\lambda(1, u)u^{\eta-1}du > 0 (\eta = \lambda - \mu, \sigma).$$

For $\mu = \lambda - \sigma_1, \hat{\sigma} = \frac{\lambda-\mu}{p} + \frac{\sigma}{q}, \hat{\mu} = \lambda - \hat{\sigma} = \frac{\lambda-\sigma}{q} + \frac{\mu}{p}$, replacing $||x||_\alpha^{\lambda_1\lambda}f(x)$ to $f(x)$, by calculation, we have

**Corollary 1.** If there exists $M(\geq 0)$, such that the following inequality holds:

$$\int_{R_+^{j_0}} \int_{R_+^{i_0}} k_\lambda(||x||_\alpha^{\lambda_1}, ||y||_\beta^{\lambda_2}) f(x)g(y)dxdy$$
$$\leq M[\int_{R_+^{i_0}} ||x||_\alpha^{p(i_0-\lambda_1\mu)-i_0} f^p(x)dx]^{\frac{1}{p}} [\int_{R_+^{j_0}} ||y||_\beta^{q(j_0-\lambda_2\sigma)-j_0} g^q(y)dy]^{\frac{1}{q}}, \tag{26}$$

then we have $\mu + \sigma = \lambda$, and $(L_\beta^{(j_0)} \frac{1}{|\lambda_2|})^{\frac{1}{p}} (L_\alpha^{(i_0)} \frac{1}{|\lambda_1|})^{\frac{1}{q}} K_\lambda(\sigma) (\in (0, M])$ is the best possible constant in (26) (for $\mu + \sigma = \lambda$).

**Corollary 2.** For $K_\lambda(\eta) \in R_+ (\eta \in \{\sigma, \lambda - \mu\})$, the following statements are equivalent:

(I) Both $(K_\lambda(\lambda - \mu))^{\frac{1}{p}} (K_\lambda(\sigma))^{\frac{1}{q}}$ and $K_\lambda(\frac{\lambda - \mu}{p} + \frac{\sigma}{q})$ are independent of $p, q$;

$$(\text{II}) \ (K_\lambda(\lambda - \mu))^{\frac{1}{p}} (K_\lambda(\sigma))^{\frac{1}{q}} \leq K_\lambda(\frac{\lambda - \mu}{p} + \frac{\sigma}{q}); \tag{27}$$

(III) $\mu + \sigma = \lambda$;

(IV) the constant factor $(L_\beta^{(j_0)} \frac{K_\lambda(\lambda - \mu)}{|\lambda_2|})^{\frac{1}{p}} (L_\alpha^{(i_0)} \frac{K_\lambda(\sigma)}{|\lambda_1|})^{\frac{1}{q}}$ in the following inequality

$$\begin{aligned}
&\int_{R_+^{j_0}} \int_{R_+^{i_0}} k_\lambda(||x||_\alpha^{\lambda_1}, ||y||_\beta^{\lambda_2}) f(x) g(y) dx dy \\
&< (L_\beta^{(j_0)} \frac{K_\lambda(\lambda - \mu)}{|\lambda_2|})^{\frac{1}{p}} (L_\alpha^{(i_0)} \frac{K_\lambda(\sigma)}{|\lambda_1|})^{\frac{1}{q}} \\
&\times [\int_{R_+^{i_0}} ||x||_\alpha^{p(i_0 - \lambda_1 \hat\mu) - i_0} f^p(x) dx]^{\frac{1}{p}} [\int_{R_+^{j_0}} ||y||_\beta^{q(j_0 - \lambda_2 \hat\sigma) - j_0} g^q(y) dy]^{\frac{1}{q}},
\end{aligned} \tag{28}$$

is the best possible;

(V) there exists a constant $M$, such that inequality (26) holds.

**Example 1.** Setting $h(u) = k(1, u)\lambda = \frac{1}{(1+u)^\lambda} (\lambda > 0; u > 0)$, we find

$$K(\eta) = K(\eta)\lambda = \int_0^\infty \frac{u^{\eta - 1}}{(1+u)^\lambda} du = B(\eta, \lambda - \eta) \in R_+ (0 < \eta < \lambda)$$

$$H(||x||_\alpha^{\lambda_1} ||y||_\beta^{\lambda_2}) = \frac{1}{(1 + ||x||_\alpha^{\lambda_1} ||y||_\beta^{\lambda_2})^\lambda}, k_\lambda(||x||_\alpha^{\lambda_1}, ||y||_\beta^{\lambda_2}) = \frac{1}{(||x||_\alpha^{\lambda_1} + ||y||_\beta^{\lambda_2})^\lambda}.$$

**Example 2.** (i) For $\lambda, \gamma > 0$, we set $H(u) = k_\lambda(1, u) = \frac{1 - u^\gamma}{1 - u^{\lambda + \gamma}} (u > 0)$. We find

$$H(||x||_\alpha^{\lambda_1} ||y||_\beta^{\lambda_2}) = \frac{1 - ||x||_\alpha^{\gamma \lambda_1} ||y||_\beta^{\gamma \lambda_2}}{1 - ||x||_\alpha^{(\lambda + \gamma)\lambda_1} ||y||_\beta^{(\lambda + \gamma)\lambda_2}}, k_\lambda(||x||_\alpha^{\lambda_1}, ||y||_\beta^{\lambda_2}) = \frac{||x||_\alpha^{\gamma \lambda_1} - ||y||_\beta^{\gamma \lambda_2}}{||x||_\alpha^{(\lambda + \gamma)\lambda_1} - ||y||_\beta^{(\lambda + \gamma)\lambda_2}}.$$

In particular, for $\gamma = \lambda$, we have

$$H(||x||_\alpha^{\lambda_1} ||y||_\beta^{\lambda_2}) = \frac{1}{1 + ||x||_\alpha^{\lambda \lambda_1} ||y||_\beta^{\lambda \lambda_2}}, k_\lambda(||x||_\alpha^{\lambda_1}, ||y||_\beta^{\lambda_2}) = \frac{1}{||x||_\alpha^{\lambda \lambda_1} + ||y||_\beta^{\lambda \lambda_2}}.$$

(ii) In view of (cf. [17]):

$$\cot x = \frac{1}{x} + \sum_{k=1}^\infty \left( \frac{1}{x - \pi k} + \frac{1}{x + \pi k} \right) (x \in (0, \pi)),$$

for $b \in (0, 1)$, by Lebesgue term by term theorem (cf. [14], we find

$$\begin{aligned}
A_b &:= \int_0^\infty \frac{u^{b-1}}{1-u} du = \int_0^1 \frac{u^{b-1}}{1-u} du + \int_1^\infty \frac{u^{b-1}}{1-u} du \\
&= \int_0^1 \frac{u^{b-1}}{1-u} du - \int_0^1 \frac{v^{-b}}{1-v} dv = \int_0^1 \frac{u^{b-1} - u^{-b}}{1-u} du \\
&= \int_0^1 \sum_{k=0}^\infty (u^{k+b-1} - u^{k-b}) du = \sum_{k=0}^\infty \int_0^1 (u^{k+b-1} - u^{k-b}) du \\
&= \sum_{k=0}^\infty \left( \frac{1}{k+b} - \frac{1}{k+1-b} \right) = \pi[\frac{1}{\pi b} + \sum_{k=1}^\infty \left( \frac{1}{\pi b - \pi k} + \frac{1}{\pi b + \pi k} \right)] \\
&= \pi \cot \pi b \in R := (-\infty, \infty).
\end{aligned}$$

**Note .** For $b \in (0, \frac{1}{2})$, $A_b > 0$; for $b \in (\frac{1}{2}, 1)$, $A_b < 0$; $A_{1/2} = 0$.

(iii) For $\eta \in (0, \lambda)$, by (ii), we obtain (cf. [18])

$$K(\eta) = K_\lambda(\eta) := \int_0^\infty H(u) u^{\eta-1} du = \int_0^\infty \frac{1-u^\gamma}{1-u^{\lambda+\gamma}} u^{\eta-1} du$$

$$\overset{v=u^{\lambda+\gamma}}{=} \frac{1}{\lambda+\gamma} \left( \int_0^\infty \frac{v^{\frac{\eta}{\lambda+\gamma}-1}}{1-v} dv - \int_0^\infty \frac{v^{\frac{\eta+\gamma}{\lambda+\gamma}-1}}{1-v} dv \right)$$

$$= \frac{\pi}{\lambda+\gamma} \left[ \cot(\frac{\pi\eta}{\lambda+\gamma}) - \cot(\frac{\pi(\eta+\gamma)}{\lambda+\gamma}) \right]$$

$$= \frac{\pi}{\lambda+\gamma} \left[ \cot(\frac{\pi\eta}{\lambda+\gamma}) + \cot(\frac{\pi(\lambda-\eta)}{\lambda+\gamma}) \right] \in R_+.$$

In particular, for $\gamma = \lambda$, we obtain

$$K(\eta) = K_\lambda(\eta) = \frac{\pi}{2\lambda} \left[ \cot(\frac{\pi\eta}{2\lambda}) + \cot(\frac{\pi(\lambda-\eta)}{2\lambda}) \right] = \frac{\pi}{\lambda \sin(\frac{\pi\eta}{\lambda})}.$$

We can use Examples 1 and 2 as the particular kernels to Theorems 1 and 2 and Corollaries 1 and 2.

## 4. Conclusions

In this article, following the idea of [7,8], by means of the technique of real analysis, the way of introduced parameters, and a few useful formulas, two new multidimensional Hilbert-type integral inequalities with the nonhomogeneous kernel as

$$H(||x||_\alpha^{\lambda_1} ||y||_\beta^{\lambda 2})(\lambda_1, \lambda \neq 20)$$

are given in (19) and (24), which are some new extensions of the Hilbert-type integral inequalities in the two-dimensional case. Some equivalent statements related to the two inequalities, the best value and several parameters are provided in Theorem 2. Two corollaries about the homogeneous kernel as $k_\lambda(||x||_\alpha^{\lambda_1}, ||y||_\beta^{\lambda 2})(\lambda_1, \lambda \neq 20)$ are given in Corollaries 1 and 2, and some new inequalities in particular parameters are obtained in Examples 1 and 2.

**Author Contributions:** Investigation, Y.L.; Writing—original draft, B.Y.; Funding acquisition, Y.Z. All authors have read and agreed to the published version of the manuscript.

**Funding:** This work is supported by the National Natural Science Foundation of China (No. 62166011) and the Innovation Key Project of Guangxi Province (No. 222068071). We are grateful for this help.

**Data Availability Statement:** No data were used to support this study.

**Conflicts of Interest:** The authors declare that they have no conflicts of interest.

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
