# Peer review of "Equivalent Statements of Two Multidimensional Hilbert-Type Integral Inequalities with Parameters"

_axioms, doi:10.3390/axioms12100956_

Round 1

Reviewer 1 Report

The manuscript:

“Equivalent Statements of Two Multidimensional Hilbert-Type Integral Inequalities with Parameters”, by Y. Li, Y. Zhong and B. Yang (Ref. No.: axioms-2572093-peer-review),

contains some interesting material and may be potentially suitable for publication. However, it is not well-organized and requires a significant elaboration. In particular, the objective and the novelty of this work is vague. Second, the paper is too technical without sufficient description.

English is acceptable. The cited literature is sufficient.

Apart from this, the following should be taken into consideration:

Abstract

The Abstract does not reflect any novelty or originality of this study.

1. Introduction

1) The sentence: “In this paper, following the idea of [7, 8], by means of the weight functions, the way of introduced parameters and the transfer formulas, two multidimensional Hilbert-type integral inequalities with the nonhomogeneous kernel as …”. Is this idea new or just a specific case that were previously suggested by Krnic et al. [7] and Adiyasuren et al. [8]? These details should be shown in the Abstract.

2. Some lemmas

1) This section shows proofs of five lemmas. What are the significances of these lemmas? More descriptions should be provided.

2) Only Lemma 5 is required for the next section ‘3 Main results and two corollaries’. Why all five lemmas are shown. Would not be enough to show just only one lemma 5?

3. Main results and two corollaries

1) This section provides the proofs for the Theorems 1 and 2. It also shows two Corollaries and a few examples. While authors devoted this work completely to technical parts in proving theorems, the description of their results are not given at all. The reader will nor be able to understand the novelty/originality of this work and its significance. More relevant descriptions signifying novel ideas in this study, should be shown.

4. Conclusion

1) Similar to Abstract, the novelty if this work should be stronger emphasized.

2) The sentence: “The further study is to consider a new multidimensional half-discrete Hilbert-type inequality involving one derivative function of higher-order by using the way of this paper and [14]”. I suggest, this sentence should be omitted. I see no reason to publicise the ideas without applications. This is also in authors interest not to publicise it.

The manuscript requires a major mandatory revision.

Reviewer 2 Report

Grammar and punctuation must be revised.

Reviewer 3 Report

The authors gave two multidimensional Hilbert-type integral inequalities with the general nonhomogeneous kernel by means of the weight functions using the idea of introduced parameters and the transfer formulas. They provided some equivalent statements of the best possible constant factor and several parameters related to the new inequalities. 

The paper is well organized and contains new findings. It can be accepted after revising the  following points.   

1.      In page 1, line 28, in the sentence  “Among them, the papers [2]… “  papers should be singular.

2.      In page 2, line 11, in the sentence  “ … and the papers [12]… “  papers should be singular.

3.      In page 2, before the sentence “Two functions..”  put dot instead of comma.

4.      In page 2, line 13, in the sentence  “ … upper limit function as a application… “  ‘a application’ should be replaced by ‘an application’.

5.      In page 2, in the last sentence, “measurable functions” should be replaced by “measurable function”

6.      In page 3, put dot before equation (4).

7.      In page 4, put dot before equation (12).

8.      In page 7 and page 8, in the word “Hölders ” change the style of the letter “ö” as the other letters.

9.      In page 7, after equation (20), “constant A and B” sould be replaced by “constants A and B”

10.  In conclusion, line 7, in the sentence “…inequalities are obtained toin”, delete the word “toin”.

The paper can be accepted after the minor revisions mentioned above.

needs minor revision which is mentioned in the attached file

Author Response

Please fsee the attached file.

Round 2

Reviewer 1 Report

The manuscript:

“Equivalent Statements of Two Multidimensional Hilbert-Type Integral Inequalities with Parameters”, by Y. Li, Y. Zhong and B. Yang (Ref. No.: axioms-2572093-peer-review-v2),

has been improved after major revision. However, it still needs some improvements and the following minor amendments may be recommended:

1) The sentence: “In this article, following the idea of [7, 8], by means of real analysis, the way of introduced parameters and the transfer formulas, two new multidimensional Hilbert-type integral inequalities with the nonhomogeneous kernel …”. This will emphasize the novelty of this work.

2) The sentence in the section ‘5 Conclusion’: “The further study is to consider a new multidimensional half-discrete Hilbert-type inequality involving one derivative function of higher-order by using the way of this paper and [14]”. The authors either should omit this sentence or at least briefly discuss about it in the section: “3 Main results and two corollaries”.

The manuscript requires a minor revision.

Author Response

Please see the attached file with the red Notes.

Reviewer 2 Report

The coverletter I have access to does not answer my remarks from the previous report. It refers to the other reviewer only. This is offending me.

The author partially did the revision I have asked for, without answering my report and without explaining the reason of ignoring some of them. No reviewer remark must be ignored.

I expect a complete revision and an answer to my report.

The authors oftenly use informal English (see, for example, the phrase following formula (20). Also, there are grammar mistakes.

Round 3

Reviewer 2 Report

The authors did revise the paper according to our remarks from the previous review.

We recommend the paper to be accepted for publication as it is now.